# National representative analysis of unilateral hearing loss and hearing aid usage in South Korea

**Se A. Lee[1], Hyun Tag Kang[2], Yun Ji Lee[2], Jong Dae Lee[2], Bo Gyung Kim[2]***

**1** Department of Otorhinolaryngology, Yonsei University College of Medicine, Seoul, Korea, **2** Department of Otorhinolaryngology-Head and Neck Surgery, Soonchunhyang University College of Medicine, Bucheon, Korea

* bgkim@schmc.ac.kr

## Abstract

A definitive study on the prevalence of adult unilateral hearing loss and hearing aid rehabilitation is lacking in Korea. The purpose of our study was to investigate the prevalence of adult unilateral hearing loss and the factors associated with hearing aid use in patients with unilateral hearing loss in South Korea. We obtained data from 2009 to 2012 from the Korea National Health and Nutrition Examination Surveys (KNHANES), a cross-sectional, nationwide and population-based survey in the Republic of Korea. We analyzed the prevalence and associated factors of unilateral hearing loss and hearing aid adoption by univariable and multivariable analysis. Unilateral hearing loss was defined as pure tone average $\geq$ 41 dB in the worse hearing ear, and < 41 dB in the other ear assessed at 0.5, 1.0, 2.0, and 3.0 kHz. From 2009 to 2012, 33,252 individuals participated in the KNHANES. Among them, the number of patients with unilateral hearing loss was 1632 (5.55%) and the prevalence of hearing aid adoption in unilateral hearing loss was 1.56%. We also compared the factors between hearing aid users and non-users. Occupational status (OR 3.759, 95% CI 1.443–9.804), the hearing threshold in the better ear (OR 1.088, 95% CI 1.029–1.151), and hearing threshold in the worse ear (OR 1.031, 1.005–1.058) were found to affect the adoption of hearing aids. The prevalence of noise exposure at work in hearing aid users was significantly lower than the prevalence of noise exposure at work in those with no hearing aid. The prevalence of hearing aid use in patients with unilateral hearing loss in Korea is very low compared to other countries. Public health education is needed to increase public awareness of unilateral hearing loss, hearing aid adoption and its continued usage. Auditory rehabilitation should be actively recommended to patients with unilateral hearing loss.

**Data Availability Statement:** All data are available from the fifth Korea National Health and Nutrition Examination Survey. The data can be obtained at the following link: knhanes.cdc.go.kr.

## Introduction

Sensorineural hearing loss (SNHL) affects approximately 1 in 500 newborns [1]. Unilateral hearing loss (UHL) is often not detected until early grade school and affects approximately

**Funding:** This work was supported by the Basic Science Research Program through the National Research Foundation of Korea, funded by the Ministry of Education, Science and Technology (2016R1D1A1B03934946), and the Soonchunhyang University Research Fund. The funders had no role in study design, data collection and analysis, decision to publish, or preparation of the manuscript.

**Competing interests:** The authors have declared that no competing interests exist.

0.1%–3.0% of children [2]. The true incidence of UHL has been difficult to determine because many patients with UHL exhibit normal speech and language development [3, 4].

Bilateral hearing loss that occurs with aging is known to be associated with dementia and falls [5, 6]. Little is known about associations between morbidity and UHL. Recent studies suggest that in children UHL is associated with cognitive impairment and poor school performance [7, 8]. Lieu [7] reported that in school-age children UHL was associated with an increased likelihood of grade failures, and that speech and language delays may occur. A meta-analysis performed by Purcell et al. [8] suggested that children with UHL have lower full-scale IQ and performance IQ than children with normal hearing. These studies indicate the importance of auditory rehabilitation in the deficient ear in children with UHL.

Auditory deprivation is one reason why patients with UHL use hearing aids. Auditory deprivation can progress systematically over time and is associated with the reduced availability of acoustic information [9]. Many studies suggest that unilateral deprivation during early development can reorganize the central auditory representation of the two ears, resulting in a stronger representation of the better hearing ear and weaker representation of the other ear. These changes lead to a persistent aural preference for one ear, as demonstrated by asymmetric speech comprehension when each ear is tested separately [10–12]. Therefore, hearing aid use is important in patients with UHL. Notably, however, no study has investigated the prevalence of UHL and hearing aid adoption and use in South Korea. Most previous epidemiologic studies of UHL have focused on children. Golub et al. [13] recently analyzed big data derived from the USA and reported that the prevalence of UHL in that country was 7.2% but only 2.0% of Americans with UHL were using hearing aids. The Korea National Health and Nutrition Examination Survey (KNHANES) makes big data available for investigating diseases and aspects of health and nutrition. In the present study, the prevalence of UHL in participants aged ≥ 12 years in South Korea was investigated, as were factors associated with hearing aid use in South Koreans with UHL.

## Materials and methods

### KNHANES and the study population

Data were obtained from the 2009–2012 KNHANES and thus the study involved the secondary analysis of a large data-set. The KNHANES is a cross-sectional, nationwide, population-based survey designed to assess national health and nutrition levels accurately that has been conducted by the Division of Chronic Disease Surveillance under the Korea Centers for Disease Control and Prevention since 1998. It is composed of three different components; a health interview, a physical examination, and a nutrition survey. Every year 10,000–12,000 individuals in approximately 4000 households are selected from a panel to represent the population using a multistage clustered and stratified random sampling method that is based on Korean national census data. Since the Korean Society of Otolaryngology-Head and Neck Surgery began to participate in the survey in 2008, it has included otolaryngologic interviews and examinations performed by well-trained otolaryngologists in a mobile unit containing an endoscopic system and audio booth. Survey sample weights are used in all analyses to generate estimates that are representative of the non-institutionalized civilian Korean population. A detailed description of the KNHANES data collection methods has been published previously [14, 15]. All participants in the KNHANES provided written informed consent prior to undertaking the survey. The present study was approved by the relevant institutional review board (file ID 2019-09-024), and that board waived the need for informed consent due to the retrospective nature of the study and the lack of any personally identifiable information in the study.

### Hearing survey and otologic examination

Pure tone audiograms were conducted in participants aged $\geq$ 12 years who were eligible for the survey. Only air conduction thresholds were measured. Data from the otolaryngologic survey and examinations were evaluated, including tympanic membrane perforation, hearing loss, dizziness, and vestibular dysfunction. Potential associations between factors derived from the basic health examination and interview and UHL and hearing aid use were assessed. To determine the prevalence of tympanic membrane perforation and cholesteatoma, as well as the retraction pocket and otitis media with effusion, an ear examination was conducted with a 4-mm 0-degree angled rigid endoscope attached to a charge-coupled device camera in all participants. Pure tone audiograms were measured by a trained otolaryngologist using an automatic audiometer (GSI SA-203; Entomed Diagnostics AB, Lena Nodin, Sweden) in a soundproof booth. Pure tone thresholds were obtained independently at 6 frequencies in each ear; 0.5, 1.0, 2.0, 3.0, 4.0, and 6.0 kHz. Pure-tone average (PTA) was calculated as the average threshold at 0.5, 1.0, 2.0, and 3.0 kHz in accordance with the recommendations of the American Academy of Otolaryngology-Head and Neck Surgery [16]. UHL was defined as a PTA hearing level $\geq$ 41 dB in one ear and $<$ 41 dB in the other. UHL was deemed to be not present if the PTA was $\geq$ 41 dB in both ears. With regard to their use of hearing aids participants were asked "Do you use hearing aid(s)?", and the response options were "yes", "yes, but rarely", and "no". Hearing aid adoption was deemed to be present in subjects who answered "yes" or "yes, but rarely", which means currently active or rarely used, respectively. Hearing aid use was deemed to be present in subjects who answered "yes", which means currently used actively.

### Statistical analysis

The prevalence of UHL was calculated and reported as a weighted percentage with standard error. In the univariable analysis, the Rao-Scott Chi-square test and logistic regression analysis were used to test associations between UHL and potentially related factors in complex sampling design. In multivariable analysis, adjusted odds ratios (ORs) with 95% confidence intervals (CIs) were calculated via logistic regression analysis. To reflect the national population estimates, sample weights were applied in all analyses. Statistical analysis was performed using SAS version 9.4 (SAS Institute, Cary, NC, USA). All $p$ values were two-sided, and $p < 0.05$ was deemed to indicate statistical significance.

## Results

### Prevalence of UHL and hearing aid adoption

A total of 33,252 individuals participated in the KNHANES from 2009–2012. Of these, the number of participants with UHL was 1,632 (weighted percentage of 5.55%, standard error 0.23). PTA $\geq$ 41 dB and $<$ 55 dB, PTA $\geq$ 55 dB and $<$ 70 dB, and PTA $\geq$ 70dB accounted for 56.86%, 21.89%, and 21.25% respectively. The prevalence of UHL in the eight different age groups are shown in Table 1. In the Rao-Scott Chi-Square test, there was a significant correlation between age group and hearing level in participants with UHL ($p < 0.0001$).

In participants with UHL, the prevalence of hearing aid adoption was 1.56% (standard error 0.37). Bilateral PTAs in hearing aid adopters with UHL are shown in Table 2. There was no hearing aid adoption when the hearing threshold in the better ear was $<$ 20 dB, and hearing aids were adopted when the hearing threshold in the better ear was $>$ 20 dB. Hearing aid adoption in the worse ear was more common in participants with a hearing threshold in the better ear of 30–40 dB than it was in participants with a hearing threshold in the better ear of 20–30 dB.

**Table 1. Prevalence of unilateral hearing loss in South Korea by age group in participants over 12 years old (n = 1623).** Values expressed as a weighted percentage (standard errors).

| | PTA results | | | |
|---|---|---|---|---|
| | ≥41 dB, <55 dB | ≥55 dB, <70 dB | ≥70 dB | overall |
| Age (years) | | | | |
| 12–19 | 1.29 (0.48) | 1.27 (0.58) | 7.50 (1.87) | 2.60 (0.58) |
| 20–29 | 4.51 (1.26) | 5.10 (1.99) | 4.78 (1.74) | 4.70 (0.89) |
| 30–39 | 5.08 (1.06) | 6.31 (2.02) | 12.88 (3.05) | 7.01 (1.15) |
| 40–49 | 8.48 (1.27) | 10.88 (2.68) | 17.57 (3.04) | 10.94 (1.15) |
| 50–59 | 21.82 (1.86) | 24.88 (3.10) | 21.83 (3.02) | 22.49 (1.40) |
| 60–69 | 25.33 (1.84) | 24.17 (2.70) | 18.49 (2.45) | 23.62 (1.35) |
| 70–79 | 26.60 (1.81) | 20.81 (2.55) | 14.22 (2.16) | 22.71 (1.37) |
| 80- | 6.74 (1.33) | 6.89 (1.50) | 2.72 (0.86) | 5.94 (0.82) |
| overall | 56.86 (1.87) | 21.89 (1.38) | 21.25 (1.74) | |

UHL, unilateral hearing loss defined as pure tone average ≥ 41 dB in the worse hearing ear, and < 41 dB in the other ear assessed at 0.5, 1.0, 2.0, and 3.0 kHz

## Factors associated with hearing aid adoption

Univariable and multivariable logistic regression analyses were performed to identify factors associated with hearing aid adoption. In univariable analysis occupational status (OR 3.759, 95% CI 1.443–9.804), the hearing threshold in the better ear (OR 1.088, 95% CI 1.029–1.151), and hearing threshold in the worse ear (OR 1.031, 1.005–1.058) were significantly associated with hearing aid adoption. In multivariable logistic regression analyses occupational status (OR 3.750, 95% CI 1.441–9.763), hearing threshold in the better ear (OR 1.088, 95% CI 1.029–1.151), and hearing threshold in the worse ear (OR 1.031, 95% CI 1.005–1.058) were significantly associated with hearing aid adoption (Table 3). There were no calibration parameters other than those shown in the table.

## Factors associated with hearing aid use

The prevalence of hearing aid use was 0.86% (standard error 0.28) in participants with UHL, and it was 55.31% (standard error 11.97) in participants who reported initial hearing aid adoption. To identify factors associated with hearing aid use, *t*-tests and Chi-square tests were performed to compare hearing aid users and non-users. In that analysis noise exposure at work was associated with hearing aid use (Table 4). The prevalence of noise exposure at work in the hearing aid users was 3.61%, which was significantly lower than the prevalence of noise

**Table 2. Bilateral pure-tone average of hearing aid adopters with unilateral hearing loss.** Values expressed as a weighted percentage (standard errors).

| PTA of better ear | PTA of worse ear | | |
|---|---|---|---|
| | ≥ 41 dB, < 55 dB | ≥ 55 dB, < 70 dB | ≥ 70 dB |
| ≤ 20dB | 0 | 0 | 0 |
| > 20dB, ≤ 30 dB | 13.78 (9.90) | 62.22 (22.99) | 44.20 (18.36) |
| > 30 dB, ≤ 40 dB | 86.22 (9.90) | 37.78 (22.99) | 55.80 (18.36) |

UHL, unilateral hearing loss defined as pure tone average ≥ 41 dB in the worse hearing ear, and < 41 dB in the other ear assessed at 0.5, 1.0, 2.0, and 3.0 kHz

**Table 3. Univariable and multivariable analyses of factors associated with hearing aid adoption with unilateral hearing loss.**

| Variables | Categories | Univariable analysis | | Multivariable analysis | |
|---|---|---|---|---|---|
| | | Odds Ratio (95% CI) | *p*-value | Odds Ratio (95% CI) | *p*-value |
| Age | | 1.005 (0.966, 1.046) | 0.8075 | | |
| Sex | Male | ref | - | | |
| | Female | 0.506 (0.192, 1.334) | 0.1683 | | |
| Income | Lower | ref | - | | |
| | Lower middle | 0.982 (0.247, 3.899) | 0.8592 | | |
| | Upper middle | 1.042 (0.279, 3.894) | 0.9602 | | |
| | Upper | 1.258 (0.346, 4.575) | 0.7001 | | |
| Education level | <Elementary school | ref | - | | |
| | Middle school | 0.644 (0.183, 2.265) | 0.8268 | | |
| | High school | 1.519 (0.507, 4.551) | 0.0891 | | |
| | College or higher | 0.270 (0.033, 2.189) | 0.2101 | | |
| Occupational status | No | ref | - | ref | - |
| | Yes | 3.433 (1.402, 8.409) | 0.0070* | 3.750 (1.441, 9.763) | 0.0068* |
| Marriage status | Yes | ref | - | | |
| | No | 2.532 (0.506, 12.669) | 0.2583 | | |
| Difficulty in hearing | No | ref | - | | |
| | Yes | . | . | | |
| Tinnitus | Yes | ref | - | | |
| | No | 0.856 (0.327, 2.240) | 0.7520 | | |
| Anxiety about tinnitus | No | ref | - | | |
| | Yes | 2.813 (0.623, 12.695) | 0.1785 | | |
| Noise exposure during work | No | ref | - | | |
| | Yes | 1.000 (0.222, 4.496) | >.9999 | | |
| TM perforation in worse ear | No | ref | - | | |
| | Yes | 0.754 (0.128, 4.454) | 0.7555 | | |
| Cholesteatoma in worse ear | No | ref | - | | |
| | Yes | 1.012 (0.172, 5.934) | 0.9897 | | |
| Otitis media with effusion in worse ear | No | ref | - | | |
| | Yes | < .001 (< .001, < .001) | < .0001 | | |
| Chronic otitis media | No | ref | - | | |
| | Yes | 1.125 (0.392, 3.228) | 0.8260 | | |
| Dizziness | No | ref | - | | |
| | Yes | < .001 (< .001, < .001) | < .0001 | | |
| Perception of stress | No | ref | - | | |
| | Yes | 0.569 (0.171,1.900) | 0.3596 | | |
| Depression mood ≥ 2 weeks | No | ref | - | | |
| | Yes | 0.655 (0.156, 2.754) | 0.5634 | | |
| Smoking | No | ref | - | | |
| | Yes | 1.724 (0.633, 4.480) | 0.2640 | | |
| Hearing threshold in better ear | | 1.088 (1.033, 1.146) | 0.0014* | 1.088 (1.029, 1.151) | 0.0031* |
| Hearing threshold in worse ear | | 1.031 (1.006, 1.056) | 0.0140* | 1.031 (1.005, 1.058) | 0.0195* |

UHL, unilateral hearing loss defined as pure tone average ≥ 41 dB in one ear, and < 41 dB in the other ear assessed at 0.5, 1.0, 2.0, and 3.0 kHz

The independent variables of the multivariable logistic regression model included the occupational status, the hearing threshold in the better ear, and the hearing threshold in the worse ear.

**Table 4. Comparison between hearing aid users and non-users with unilateral hearing loss.** Values expressed as weighted percentage (standard errors).

| Variables | Categories | Uilateral hearing loss | Non-users | Users | *p*-value |
|---|---|---|---|---|---|
| Age | | 58.14 (0.77) | 64.41 (7.26) | 57.62 (6.64) | 0.3496 |
| Sex | Male | 51.35 (1.51) | 80.71 (12.10) | 55.59 (16.22) | 0.2142 |
| | Female | 48.65 (1.51) | 19.29 (12.10) | 44.41 (16.22) | |
| Income | Lower | 32.40 (1.56) | 30.15 (16.49) | 32.53 (15.55) | 0.3353 |
| | Lower middle | 24.92 (1.51) | 16.46 (10.18) | 28.80 (16.77) | |
| | Upper middle | 21.76 (1.62) | 11.33 (10.73) | 27.36 (13.90) | |
| | Upper | 20.91 (1.55) | 42.06 (18.23) | 11.31 (6.98) | |
| Education level | <Elementary school | 43.30 (1.78) | 24.25 (14.85) | 62.50 (16.54) | - |
| | Middle school | 15.54 (1.08) | 13.02 (9.27) | 8.06 (5.99) | |
| | High school | 28.21 (1.56) | 62.73 (16.39) | 24.22 (16.85) | |
| | College or higher | 12.95 (1.30) | - | 5.22 (5.22) | |
| Occupation | No | 53.62 (1.68) | 78.20 (11.59) | 77.39 (10.95) | 0.9586 |
| | Yes | 46.38 (1.68) | 21.80 (11.59) | 22.61 (10.95) | |
| Marriage status | Yes | 91.03 (1.28) | 100.00 (-) | 70.95 (16.74) | - |
| | No | 8.97 (1.28) | - | 29.05 (16.74) | |
| Difficulty in hearing | No | 54.98 (1.95) | 100.00 (-) | 100.00 (-) | - |
| | Yes | 45.02 (1.95) | . | . | |
| Tinnitus | Yes | 35.37 (1.58) | 57.59 (16.94) | 47.52 (16.39) | 0.6638 |
| | No | 64.63 (1.58) | 42.41 (16.94) | 52.48 (16.39) | |
| Anxiety about tinnitus | No | 54.96 (2.82) | 13.45 (13.30) | 38.57 (23.13) | 0.2845 |
| | Yes | 45.04 (2.82) | 86.55 (13.30) | 61.43 (23.13) | |
| Noise exposure during work | No | 13.78 (1.23) | 27.93 (17.72) | 3.61 (3.73) | 0.0240* |
| | Yes | 86.22 (1.23) | 72.08 (17.72) | 96.39 (3.73) | |
| TM perforation in worse ear | No | 39.43 (4.58) | . | 76.66 (21.73) | - |
| | Yes | 60.57 (4.58) | 100 (-) | 23.34 (21.73) | |
| Cholesteatoma in worse ear | No | 58.51 (4.50) | 100 (-) | 23.34 (21.73) | - |
| | Yes | 41.49 (4.50) | - | 76.66 (21.73) | |
| Otitis media with effusion in worse ear | No | 90.07 (2.35) | 100.00 (-) | 100.00 (-) | - |
| | Yes | 9.93 (2.35) | . | . | |
| Chronic otitis media | No | 82.31 (1.47) | 74.89 (13.71) | 70.24 (15.49) | 0.8178 |
| | Yes | 17.69 (1.47) | 25.11 (13.71) | 29.76 (15.49) | |
| Dizziness | No | 92.50 (2.27) | 100.00 (-) | 100.00 (-) | - |
| | Yes | 7.50 (2.27) | . | . | |
| Perception of stress | No | 74.31 (1.47) | 92.24 (7.66) | 68.66 (15.99) | 0.1458 |
| | Yes | 25.69 (1.47) | 7.76 (7.66) | 31.34 (15.99) | |
| Depression mood ≥ 2 weeks | No | 82.68 (1.14) | 100.00 (-) | 72.16 (15.77) | - |
| | Yes | 17.32 (1.14) | . | 27.84 (15.77) | |
| Smoking | No | 47.82 (1.64) | 72.97 (13.90) | 53.34 (17.01) | 0.3703 |
| | Yes | 52.18 (1.64) | 27.03 (13.90) | 46.66 (17.01) | |
| Hearing threshold in better ear | | 24.58 (0.50) | 31.07 (1.15) | 35.48 (1.15) | 0.0726 |
| Hearing threshold in worse ear | | 57.39 (0.69) | 58.98 (7.06) | 64.67 (5.69) | 0.9117 |

UHL, unilateral hearing loss defined as pure tone average ≥ 41 dB in one ear, and < 41 dB in the other ear assessed at 0.5, 1.0, 2.0, and 3.0 kHz

exposure at work in the hearing aid non-users (27.93%; *p* = 0.0240). None of the other factors analyzed were statistically significant.

## Discussion

In the present study conducted in South Korea, the prevalence of UHL determined via KNHANES data was 4.91%. This is lower than the prevalence of 7.9%–13.3% in the general population [17, 18] and the prevalence of UHL was 7.2% in the US [13]. Although it was not possible to distinguish SNHL from conductive hearing loss, most patients with UHL had normal tympanic membranes and it was concluded that most of them had SNHL. However, our definition of UHL was > 40 dB, which is different from the previously cited studies, which used > 25 dB to define UHL. Thus, we cannot make direct comparisons.

In the present study, in participants with UHL, the prevalence of hearing aid adoption was 0.86% and the prevalence of hearing aid use was 1.56%. Hearing aid use prevalence of 14.2% in the USA [19] and 21.5% in the United Kingdom have been reported [20]. Those studies were based on patients aged > 50 years or 40 years with bilateral hearing loss, whereas the participants in the current study were aged ≥ 12 years. Golub et al. [13] reported that the prevalence of hearing aid use in people with UHL was 2.0% in the USA and that the reason why it was much lower than that in people with bilateral hearing loss was unawareness of disability. Public health insurance systems may also affect hearing aid use. In European countries such as the United Kingdom, France, and Denmark hearing aid-related expenses are covered by public health insurance [21]. In the USA the use of hearing aids has steadily increased, and most of this growth has been attributed to the provision of free hearing aids that can be obtained via the Department of Veterans Affairs, or the availability of low-cost hearing aids via the internet [22]. In South Korea hearing aid-related expenses are not covered by the government health insurance system however, and the average price of hearing aids is relatively high with reference to the national income per capita.

Occupational status, the hearing threshold in the better ear, and hearing threshold in the worse ear were associated with hearing aid adoption in the present study. Patients with UHL may not feel discomfort in their daily life. If they attend meetings or are required to spend time in a noisy environment however, problems can arise from UHL. Patients with UHL experience difficulty with sound localization and speech recognition in noisy environments [23]. Additionally, their co-workers may have problems communicating with them. Hearing aid use in the work-place can evidently increase productive hearing capacity and enhance the ability to communicate with others [24, 25].

People who adopt hearing aids tend to have more severe hearing loss in both the better ear and the worse ear than non-adopters. In a previous investigation, the severity of hearing loss was a major determinant of hearing aid adoption [26]. In another study people with severely restrictive hearing thresholds reported greater amounts of hearing aid use per day, and higher levels of satisfaction with their hearing aids [27]. In comparison with objectively measured severity of hearing loss, a self-perceived degree of hearing impairment was reportedly a stronger determinant of hearing aid adoption in many studies [28, 29]. In the present study, however, difficulty hearing was not statistically significantly associated with hearing aid adoption.

Noise exposure during work was the only factor that was significantly associated with hearing aid use in the present study. Hearing aid users were exposed to less noise at work than non-users. In noisy environments patients with hearing impairment can be free from noise without hearing aids. Unlike bilateral SNHL, it is thought that patients with unilateral SNHL have a lower need for hearing aids and that it is not necessary to wear hearing aids even in noisy situations. There have been recent advances in hearing aid technology. Specifically, noise reduction systems have been incorporated into hearing aids to improve the signal-to-noise ratio and listening comfort. These advances may have resulted in improved noise tolerance in hearing aid users.

The current study had some limitations. The KNHANES data it utilized were acquired from 2009–2012 and thus were ≥ 7 years old, and hearing aid technology has advanced rapidly in the last 5 years. With regard to financial considerations, in November 2015 the hearing aid subsidy provided by government health insurance in Korea was raised to KRW 1,310,000. As a result, hearing aid adoption and use may have increased since the 2009–2012 KNHANES was conducted. Second, the KNHANES defined UHL as a PTA hearing level ≥ 41 dB in one ear and < 41 dB in the other, whereas other studies defined UHL as a PTA hearing level > 25 dB. Another study limitation is that the KNHANES questionnaire components pertaining to hearing aids are not detailed, thus the scope for analyzing factors associated with the adoption and use of hearing aids was restricted. Notably, however, the present study was based on a very large national cross-sectional survey of a representative sample based on Korean national census data, and by design, the study had the capacity to reliably determine the prevalence of UHL and overall hearing aid use.

In conclusion, the prevalence of hearing aid use in patients with UHL is very low in South Korea compared to other countries. Public health education is needed in an effort to increase public awareness of UHL and hearing aid adoption and continued use, and auditory rehabilitation should be actively recommended to patients with UHL.

## Author Contributions

**Conceptualization:** Se A. Lee, Bo Gyung Kim.

**Formal analysis:** Yun Ji Lee.

**Funding acquisition:** Bo Gyung Kim.

**Investigation:** Hyun Tag Kang.

**Methodology:** Jong Dae Lee.

**Project administration:** Bo Gyung Kim.

**Supervision:** Bo Gyung Kim.

**Writing – original draft:** Se A. Lee.

**Writing – review & editing:** Se A. Lee, Bo Gyung Kim.

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
