## [Decision Letter · Decision Letter 0]

2 Jan 2020

PONE-D-19-32331

Big data analysis of unilateral hearing loss and hearing aid use in South Korea

PLOS ONE

Dear Kim,

Thank you for submitting your manuscript to PLOS ONE. After careful consideration, we feel that it has merit but does not fully meet PLOS ONE’s publication criteria as it currently stands. Therefore, we invite you to submit a revised version of the manuscript that addresses the points raised during the review process.

We would appreciate receiving your revised manuscript by Feb 16 2020 11:59PM. To enhance the reproducibility of your results, we recommend that if applicable you deposit your laboratory protocols in protocols.io, where a protocol can be assigned its own identifier (DOI) such that it can be cited independently in the future. For instructions see: http://journals.plos.org/plosone/s/submission-guidelines#loc-laboratory-protocols

We look forward to receiving your revised manuscript.

Kind regards,

Clifford R. Hume, MD PHD

Academic Editor

PLOS ONE

Journal Requirements:

Reviewers' comments:

Reviewer's Responses to Questions

**Comments to the Author**

1. Is the manuscript technically sound, and do the data support the conclusions?

Reviewer #1: Partly

Reviewer #2: Partly

2. Has the statistical analysis been performed appropriately and rigorously? 

Reviewer #1: I Don't Know

Reviewer #2: No

3. Have the authors made all data underlying the findings in their manuscript fully available?

Reviewer #1: Yes

Reviewer #2: Yes

4. Is the manuscript presented in an intelligible fashion and written in standard English?

Reviewer #1: Yes

Reviewer #2: Yes

5. Review Comments to the Author

Reviewer #1: 1. REVIEWER SUMMARY: This is a report of unilateral hearing loss prevalence and also hearing aid use in South Korea. It uses the KNHANES, which should be representative of the actual population through sample weighting. Reporting of accurate, unbiased prevalence statistics is very important. While this has been done, for example, in the US, it is important to replicate in other countries because prevalence may differ. This study contains these important data, but there are a number of limitations detailed below. For example, multivariable regression was used but there was no obvious mention of what the variables were that were adjusted for in the model, which is critical information. In addition, the definition of unilateral hearing used a 40 dB cutoff, which is different from other papers that were mentioned in the discussion. While the authors are free to chose whatever defintion they wish, they should (a) justify the definition and (b) not make comparisons to studies that use a different definition unless they explicitly discuss this.

2. UNIQUE ASPECTS: There have been a few studies on hearing loss prevalence using the KNHANES that were not mentioned. For example: https://www.ncbi.nlm.nih.gov/pubmed/25216153 and https://www.ncbi.nlm.nih.gov/pubmed/28196098 . Both, in fact, reported unilateral hearing loss, although different definitions may have been used. This should be mentioned.

3. STUDY DESIGN: Cross-sectional

4. WRITING STYLE: Mostly understandable English. Abstract needs proofreading by a native English speaker. Many wording issues, mostly in the abstract:

Line 37-38. What does "occupation presence" mean. Does that mean having an occupation (i.e., "working")?

Line 34. Do not begin a sentence with And.

Lines 40-42 "In comparison of hearing aid users and non-users, noise exposure during work in hearing aid users was significantly lower than hearing aid non-users." Needs rewording.

Lines 42-43. "The prevalence of hearing aid use in patients with unilateral hearing loss in South Korea is very low compared as other countries." Do you mean that the low use is similar to other countries? In that case, you want to write "comparable to". Otherwise it should be "compared with" or "compared to".

Line 43-45. "Public health education is needed for increased insight and auditory rehabilitation for unilateral hearing loss should be recommended actively." Needs rewording.

Lines 104-106. "To determine the prevalence of tympanic membrane perforation and cholesteatoma, including retraction pocket and otitis media with effusion" Needs rewording. Retraction pockets and otitis media with effusion are not types of tympanic membrane perforations or cholesteatoma.

5. SPECIFIC COMMENTS:

Big data is sort of a buzz word. This is a relatively big dataset, but there are many studies that use KNHANES or the US NHANES and they normally don't use the term big data, I would at least remove from the title. You could maybe replace with something like "Nationally representative" to indicate it wasn't just based on a single hospital's audiology clinic.

Line 26-27. Do you mean lacking in South Korea? There are other studies using population-level data, e.g., reference #13.

Line 48-51. You state the incidence of unilateral hearing loss and then in the next sentence state that the true incidence is not known. In between, you should explain the limitations of the existing methods, i.e., why they are not accurate.

Line 57. It is Purcell, not Percell.

Line 113-115. Your definition of PTA >= 41 dB for hearing loss is different from many of the recent American epidemiology papers which use >=26 dB. As I am familiar, 25-40 dB is often considered mild hearing loss, which is still hearing loss. How you define hearing loss is a matter of debate and certainly varies. I would place your definition of unilateral hearing loss in the abstract to make this clear (i.e., >= 40 dB PTA at 0.5, 1, 2, 3 KHz in the worse hearing ear). I would also place this definition as a footnote in all of your tables.

I would recommend not double-spacing the tables so they are not spread across four pages (i.e., Table 3).

Table 1. You present categories, but the total data. It would be helpful to have an overall column that is not divided into a severity category. i.e, >=41 dB. It would also be helpful to have an age >= 12 row to represent "any age."

Line 121. It seems you presented the prevalence of UHL with standard eror, not 95% CI (e.g., tables 1, 2, 4)

Line 170. "In multivariable logistic regression..." You have to state what the other variables that you adjusted for were! This should also be clear in Table 3, e.g., in a footnote. This should also be in the methods.

The Table title and results text (and methods) should all match and reflect what was actually done. E.g., Table 3 title states the outcome is hearing aid adoption or cochlear implantation. The results and methods text do not mention cochlear implantation. In fact, nowhere else in the paper is cochlear implant mentioned as far as I see.

I am very confused by the different method used to assess factors associated with hearing aid use versus factors associated with hearing aid adoption. Hearing aid use and hearing aid adoption are VERY similar, with the later also including people who have a hearing aid but rarely use them. Why do you use regression to look for variables associated with hearing aid adoption, but then you use T-tests or Chi-squared tests for variables associated with hearing aid use.

Lines 193-197. This is very important. If you are comparing your findings of unilateral hearing loss to other papers measuring hearing loss, your definitions must be the same! You cite ref #16 Agrawal but she used >25 dB to define hearing loss, but you used >40 dB. Those will lead to VERY different numbers! Also ref #13 Golub et al used > 25 dB. I think ref #17 Chia et al also used >25 dB.

The similarity of the terms hearing aid use and hearing aid adoption is a little confusing. In the results and the discussion, I would clarify that use means something like current active use. And adoption means something like current active or rare use.

Methods - you must define how all variables were measured. This includes all variables in the table. For example, how was noise exposure during work measured and what does it even mean? It can mean extremely loud noise such as explosions or factory noise. This could injure your hearing. It can also mean background noise like people talking. This would not injure your hearing but can make hearing with a mild hearing loss more difficult.

Line 249 "In conclusion, the prevalence of hearing aid use in patients with UHL is very low in South Korea compared to other countries." This is your key concluding statement, but I am not sure I can trust it because your definition of UHL may be different from the studies you compared to (see above)

Reviewer #2: � Line 8; This article did not focus on congenital UHL. So, introduction like this is inappropriate.

Line 63; What does “development” mean? Development of what?

Line 100; Why did you exclude children younger than 12 years-old?

Line 113-4; Was this definition of PTA suitable for UHL or asymmetric HL? General criteria for “hearing loss” is 26dBHL. 40dBHL is used as the criteria of “disabling hearing loss”, NOT “hearing loss” for adults. The criteria of “disabling hearing loss” is 30dBHL for children. Please, provide the reference of authors’ criteria for UHL. I think that authors should have used other term rather than UHL if they insisted on the 40dBHL-criteria.

Line 142-3; Authors told that they analyzed the hearing use in people with UHL, but their data concluded the hearing aid was more used in asymmetric HL. How do you think about my point-out?

To identify the factors associated with hearing aid adoption, logistic regression analyses were performed. But to identify factors associated with hearing aid use, t-tests and Chi-square tests were performed. Why did you use different statistical methods?

Table 1; What was the “weighted percentage”? On what was it weighted?

Table 2; Why was PTA average of better ear classified into {≤ 20dB}, {> 20dB, ≤ 30 dB} and {> 30 dB, ≤ 40 dB}?

Table 2; please, rewrite the data correctly.

Table 3; Cochlear implantation? What is this?

6. PLOS authors have the option to publish the peer review history of their article (what does this mean?). If published, this will include your full peer review and any attached files.

Reviewer #1: No

Reviewer #2: No

---

## [Author Response · Author response to Decision Letter 0]

9 Feb 2020

February 10, 2020

Joerg Heber

Editor

PLOS ONE

Dear Dr Heber,

Please consider our revised manuscript, “National representative analysis of unilateral hearing loss and hearing aid usage in South Korea”, for publication in PLOS One

We appreciate the interest that the editors and reviewers have taken in our manuscript and the constructive criticism they have given. We have addressed the major concerns of the reviewers. We have also included a point-by-point response to the reviewers in addition to making the changes in the manuscript. Changes to the text in the manuscript are marked in red. Our main findings remain unchanged.

This manuscript has not been published or presented elsewhere in part or in entirety and is not under consideration by another journal. The study design was approved by the appropriate ethics review board. We have read and understood your journal’s policies, and we believe that neither the manuscript nor the study violates any of these. There are no conflicts of interest to declare.

Bo Gyung Kim, MD, PhD

Department of Otorhinolaryngology-Head and Neck Surgery, Soonchunhyang University College of Medicine

Address: 170 Jomaru-ro, Wonmi-gu, Bucheon 14584, Korea

Tel: +82-32-621-6951

Fax: +82-32-621-5018

E-mail: bgkim@schmc.ac.kr

---

## [Decision Letter · Decision Letter 1]

23 Mar 2020

PONE-D-19-32331R1

National representative analysis of unilateral hearing loss and hearing aid usage in South Korea

PLOS ONE

Dear Kim,

Thank you for submitting your manuscript to PLOS ONE. After careful consideration, we feel that it has merit but does not fully meet PLOS ONE’s publication criteria as it currently stands. Therefore, we invite you to submit a revised version of the manuscript that addresses the points raised during the review process.

Please carefully review all recommendations for revision in the original submission and re-submission.  While the authors did address the comments in their response, a number of critical clarifications were not incorporated into the revised manuscript.  The reviewer for revised submission has very carefully pointed out some essential changes that clarify how the current manuscript differs in some definitions from others in the literature, and how variables in their analysis were defined in KHANES data. 

We would appreciate receiving your revised manuscript by May 07 2020 11:59PM. To enhance the reproducibility of your results, we recommend that if applicable you deposit your laboratory protocols in protocols.io, where a protocol can be assigned its own identifier (DOI) such that it can be cited independently in the future. For instructions see: http://journals.plos.org/plosone/s/submission-guidelines#loc-laboratory-protocols

We look forward to receiving your revised manuscript.

Kind regards,

Clifford R. Hume, MD PHD

Academic Editor

PLOS ONE

Reviewers' comments:

Reviewer's Responses to Questions

**Comments to the Author**

1. If the authors have adequately addressed your comments raised in a previous round of review and you feel that this manuscript is now acceptable for publication, you may indicate that here to bypass the “Comments to the Author” section, enter your conflict of interest statement in the “Confidential to Editor” section, and submit your "Accept" recommendation.

Reviewer #1: (No Response)

2. Is the manuscript technically sound, and do the data support the conclusions?

Reviewer #1: (No Response)

3. Has the statistical analysis been performed appropriately and rigorously? 

Reviewer #1: (No Response)

4. Have the authors made all data underlying the findings in their manuscript fully available?

Reviewer #1: (No Response)

5. Is the manuscript presented in an intelligible fashion and written in standard English?

Reviewer #1: (No Response)

6. Review Comments to the Author

Reviewer #1: Thank you for the revisions and the detailed responses to my comments. Several of my comments from the prior submission remain unaddressed in the manuscript. In general, when a reviewer has a comment that requires a clarification, they would like the clarification to be inside the manuscript, not just in the response to reviewer comments. If the reviewer had a question, then many readers may have the same question.

1. NEW COMMENTS. (Line numbers refer to the new submission version with tracked changes)

Line 42 "...noise exposure at work in users with no hearing aid" should be "...noise exposure at work in those with no hearing aid"

2. PRIOR COMMENTS that were not addressed by modifications to the manuscript. NOTE that line numbers here refer to the **first submission**) Your comments are denoted by >>. My new comments are denoted by >>>

Line 26-27. Do you mean lacking in South Korea? There are other studies using population- level data, e.g., reference #13.

>>Of course, several studies have been published, such as reference # 13, however, there are very few studies on the use of UHL and hearing aids in Korea as well as in other countries, compared to studies on bilateral SNHL.

>>>Thank you, I understand. Please make this more clear in the manuscript by rewording the first sentence of the abstract to add "in Korea" to the end of this current sentence: "A definitive study on the prevalence of adult unilateral hearing loss and hearing aid rehabilitation is lacking."

Line 113-115. Your definition of PTA >= 41 dB for hearing loss is different from many of the recent American epidemiology papers which use >=26 dB. As I am familiar, 25-40 dB is often considered mild hearing loss, which is still hearing loss. How you define hearing loss is a matter of debate and certainly varies. I would place your definition of unilateral hearing loss in the abstract to make this clear (i.e., >= 40 dB PTA at 0.5, 1, 2, 3 KHz in the worse hearing ear). I would also place this definition as a footnote in all of your tables.

>>Thank you for your valuable comment and suggestion. Many studies define unilateral hearing loss as PTA >= 26 dB. However, the KNHANES defined unilateral hearing loss as PTA >=41 dB. Moreover, the prevalence of unilateral hearing loss in our study is much lower than other studies, which define unilateral hearing loss as PTA >=25 dB. If we change the definition of PTA >=26dB rigidly, there would be no one who used a hearing aid.

>>>Thank you for the explanation. You justify why you use >=40 dB. I understand this. However, because this is not the worldwide standard definition of hearing loss, I recommend doing this (repeated from above): Place your definition of unilateral hearing loss in the abstract to make this clear (i.e., >= 40 dB PTA at 0.5, 1, 2, 3 KHz in the worse hearing ear). I would also place this definition as a footnote in all of your tables."

Line 170. "In multivariable logistic regression..." You have to state what the other variables that you adjusted for were! This should also be clear in Table 3, e.g., in a footnote. This should also be in the methods.

>>The independent variables of the multivariable logistic regression model are the occupational status, the hearing threshold in the better ear, and the hearing threshold in the worse ear. There are no calibration parameters other than those shown in the table. In the univariable analysis results, the significant variables were considered as independent variables of the multivariable analysis. Among the five variables that were significant in univariable analysis, otitis media and dizziness were difficult to estimate due to an insufficient number of patients.

>>>I understand. Often in multivariable regression, there are other variables that are non-significant (i.e., pontential confounders). Since all 3 variables you describe are significant, it is not clear that you did not include any other variables in the model. To make this clear, at the end of the sentence "In multiple logistic regression..." please insert another sentence and explain that there were no other variable present in the multivariable model. Please also, like I said, put this information in the methods and in a Table 3 footnote.

Lines 193-197. This is very important. If you are comparing your findings of unilateral hearing loss to other papers measuring hearing loss, your definitions must be the same! You cite ref #16 Agrawal but she used >25 dB to define hearing loss, but you used >40 dB. Those will lead to VERY different numbers! Also ref #13 Golub et al used > 25 dB. I think ref #17 Chia et al also used >25 dB.

>>As mentioned earlier, the KNHANES defined unilateral hearing loss as PTA >=41 dB. Moreover, the prevalence of unilateral hearing loss in our study is much lower than other studies, which define unilateral hearing loss as PTA >=25 dB. If we change the definition of PTA >=26dB rigidly, there would be no one who used a hearing aid. The criteria are less strict than other studies, but with lower prevalence, we believe our results are meaningful.

>>>Thank you and I understand why you used >40 dB: you had no choice because this is what KHANES used. However, please modify the text here to explain that your definition of hearing loss is different from the papers you are comparing it to. The following statement is completely misleading without an explanation: "In the present study conducted in South Korea, the prevalence of UHL determined via KNHANES data was 4.91%. This is lower than the prevalence of 7.9%–13.3% in the general population [16, 17] and the prevalence of UHL was 7.2% in the US [13]." You must insert a sentence here to say something like: "However, our definition of hearing loss was >40 dB which is different from the previously cited studies, which used >25 dB to define hearing loss. Thus we can not make direct comparisons."

Methods - you must define how all variables were measured. This includes all variables in the table. For example, how was noise exposure during work measured and what does it even mean? It can mean extremely loud noise such as explosions or factory noise. This could injure your hearing. It can also mean background noise like people talking. This would not injure your hearing but can make hearing with a mild hearing loss more difficult.

>>We used the data obtained from the 2009-2012 KNHANES. Participants were asked for the presence of noise but there was no query on the degree of noise.

>>>Thank you. Please address my comment. Please define how all the variables were measured or defined, briefly according to the KHANES manual, in the methods. If there is no room, place this in a supplement or a reference to the KHANES manual.

7. PLOS authors have the option to publish the peer review history of their article (what does this mean?). If published, this will include your full peer review and any attached files.

Reviewer #1: No

---

## [Author Response · Author response to Decision Letter 1]

7 Apr 2020

We thank you and the reviewers for your thoughtful suggestions and insights. The manuscript has benefited from these insightful suggestions. I look forward to working with you and the reviewers to move this manuscript closer to publication in Plos One.

The manuscript has been rechecked and the necessary changes have been made in accordance with the reviewers’ suggestions. This included editing the entire document for readability and organization. Additions to the manuscript are marked with red text. The responses to all comments have been prepared and attached herewith.

Thank you for your consideration. I look forward to hearing from you.

---

## [Editor Report · Decision Letter 2]

8 Apr 2020

National representative analysis of unilateral hearing loss and hearing aid usage in South Korea

PONE-D-19-32331R2

Dear Dr. Kim,

We are pleased to inform you that your manuscript has been judged scientifically suitable for publication and will be formally accepted for publication once it complies with all outstanding technical requirements.

With kind regards,

Clifford R. Hume, MD PHD

Academic Editor

PLOS ONE

---

## [Editor Report · Acceptance letter]

13 Apr 2020

PONE-D-19-32331R2 

National representative analysis of unilateral hearing loss and hearing aid usage in South Korea 

Dear Dr. Kim:

I am pleased to inform you that your manuscript has been deemed suitable for publication in PLOS ONE. Congratulations! Your manuscript is now with our production department. 

With kind regards,

on behalf of

Dr. Clifford R. Hume 

Academic Editor

PLOS ONE